# Do Pregnancy-Induced Brain Changes Reverse? The Brain of a Mother Six Years after Parturition

**DOI:** 10.3390/brainsci11020168

**Published:** 2021-01-28

**Authors:** Magdalena Martínez-García, María Paternina-Die, Erika Barba-Müller, Daniel Martín de Blas, Laura Beumala, Romina Cortizo, Cristina Pozzobon, Luis Marcos-Vidal, Alberto Fernández-Pena, Marisol Picado, Elena Belmonte-Padilla, Anna Massó-Rodriguez, Agustin Ballesteros, Manuel Desco, Óscar Vilarroya, Elseline Hoekzema, Susanna Carmona

**Affiliations:** 1Instituto de Investigación Sanitaria Gregorio Marañón, 28007 Madrid, Spain; mmartinez@hggm.es (M.M.-G.); mpaternina@hggm.es (M.P.-D.); dmartindeblas@hggm.es (D.M.d.B.); lmarcos@hggm.es (L.M.-V.); afernandez@hggm.es (A.F.-P.); desco@hggm.es (M.D.); 2Centro de Investigación Biomédica en Red de Salud Mental (CIBERSAM), 28007 Madrid, Spain; 3Institute of Mental Health Vidal i Barraquer, Ramon Llull University, 08022 Barcelona, Spain; erika.barba.muller@gmail.com; 4Departament de Psiquiatria i Medicina Legal, Universitat Autònoma de Barcelona, 08193 Cerdanyola del Vallés, Spain; laurabeumalaa@gmail.com; 5CSMA Ciutat Vella, Parc Sanitari Sant Joan de Deu, 08003 Barcelona, Spain; r.cortizo@pssjd.org; 6Instituto Valenciano de Infertilidad, 20145 Milan, Italy; cristina.pozzobon@ivirma.com; 7Departamento de Bioingeniería e Ingeniería Aeroespacial, Universidad Carlos III de Madrid, 280829 Madrid, Spain; 8Institut Hospital del Mar d’Investigacions Mèdiques, 08003 Barcelona, Spain; marisol.picado@gmail.com; 9Divisió de Salut Mental, Althaia, Xarxa Assistencial Universitària de Manresa, 08243 Manresa, Spain; ebelmontep@althaia.cat; 10Instituto de Neuropsiquiatría y Adicciones (INAD), Parc de Salut Mar, 08003 Barcelona, Spain; annamasso20@gmail.com; 11Instituto Valenciano de Infertilidad (IVI Barcelona), 08017 Barcelona, Spain; agustin.ballesteros@ivirma.com; 12Centro Nacional de Investigaciones Cardiovasculares (CNIC), 28029 Madrid, Spain; 13Brain and Development Research Center, Leiden University, 2333 Leiden, The Netherlands; e.a.hoekzema@fsw.leidenuniv.nl; 14Leiden Institute for Brain and Cognition, 2300 Leiden, The Netherlands

**Keywords:** pregnancy, maternal brain, magnetic resonance imaging, neuroplasticity, postpartum

## Abstract

Neuroimaging researchers commonly assume that the brain of a mother is comparable to that of a nulliparous woman. However, pregnancy leads to pronounced gray matter volume reductions in the mother’s brain, which have been associated with maternal attachment towards the baby. Beyond two years postpartum, no study has explored whether these brain changes are maintained or instead return to pre-pregnancy levels. The present study tested whether gray matter volume reductions detected in primiparous women are still present six years after parturition. Using data from a unique, prospective neuroimaging study, we compared the gray matter volume of 25 primiparous and 22 nulliparous women across three sessions: before conception (*n* = 25/22), during the first months of postpartum (*n* = 25/21), and at six years after parturition (*n* = 7/5). We found that most of the pregnancy-induced gray matter volume reductions persist six years after parturition (classifying women as having been pregnant or not with 91.67% of total accuracy). We also found that brain changes at six years postpartum are associated with measures of mother-to-infant attachment. These findings open the possibility that pregnancy-induced brain changes are permanent and encourage neuroimaging studies to routinely include pregnancy-related information as a relevant demographic variable.

## 1. Introduction

Motherhood is a life-changing event that affects the social, psychological, and biological spheres. Biologically, pregnancy entails dramatic adaptations in the function and structure of all physiological systems, including the brain. Studies in non-human animal models indicate that pregnancy and motherhood modify the so-called maternal circuit: a set of brain regions that includes reward and social processing areas [1,2,3]. Such modifications, triggered by pregnancy and peripartum hormonal fluctuations and then by mother–pup interactions, play a critical role in the onset, maintenance, and adjustment of maternal care [4].

Results from neuroimaging studies in humans closely align with those obtained in non-human animal models. For instance, functional magnetic resonance imaging (MRI) studies show that the mother’s brain responds to her child’s stimuli by activating regions involved in reward and social processing [5]. According to this literature, activations in reward regions, especially in the nucleus accumbens, reflect the hedonic state that the mothers experience when presented with stimuli of their child [6]. Activations in areas involved in social processing, such as medial prefrontal cortex and precuneus, reflect mentalizing or Theory of Mind (ToM) processes, that is, the disposition of a person to interpret others’ minds, in this case, the mother’s disposition to interpret her infant’s signs [7].

As compared to functional studies, very few have investigated whether pregnancy and motherhood modify brain anatomy. Existing longitudinal data indicate that the transition to motherhood also produces anatomical adaptations in reward and social cognition brain circuits [8]. These adaptations seem to be dynamic: They differ in direction and magnitude depending on the time frame studied (pregnancy vs. postpartum) [9,10,11,12]. Specifically, whereas studies comparing preconception to postpartum brains found gray matter (GM) volume decreases [13,14,15], those analyzing the brain from the early to late postpartum period found GM volume increases [13,16,17]. This suggests a U-shaped trajectory of volume decreases during pregnancy, followed by increases after parturition. However, it remains to be determined if brain volume ever fully returns to pre-pregnancy levels. The longest longitudinal study to date that tested the persistence of brain changes after the early postpartum compared the neuroanatomical MRI data of primiparous women across three sessions: a few months before their first pregnancy, during the early postpartum, and at two years after parturition [14]. In this study, we found pronounced GM volume reductions within ToM networks following the first pregnancy, and these reductions were still detectable two years after parturition.

The first two years postpartum—which together with the intrauterine period are known as “the first thousand days of life”—are considered a critical period to promote optimal child development [17]. As infants are highly dependent, this critical period of nurturance and care requires a tremendous maternal investment [18]. Since early maternal care has been associated with neuroanatomical changes during pregnancy [14] and postpartum [16,19], it is possible that mother’s brain structure returns to pre-pregnancy levels after these first two critical years. If changes remit, they might indicate that they are confined to the period of maximal maternal investment and that, after that, the mother’s brain is equivalent to that of nulliparous women. On the contrary, if changes are still detectable after this critical period, they might indicate that pregnancy has a long-lasting—perhaps irreversible—impact on women’s brains. Still, it is unknown how the anatomy of a woman’s brain evolves beyond the first two years postpartum.

Here, we assessed if a single pregnancy leaves a trace in the anatomy of a woman’s brain that persists beyond the first two years of maximal maternal investment. For that, we followed the subjects of the prospective cohort study of Hoekzema et al., 2017 [14], and scanned them again at six years after parturition. We compared the brain of primiparous and nulliparous women across three sessions: before conception (PRE), during the early postpartum (POST), and six years after parturition (POST6y). We aimed to determine: (1) if the GM volume reductions detected in the mother’s brain during their first and only pregnancy were still present six years after parturition and (2) if, based on GM changes, we could classify women as never having been pregnant or as having been pregnant more than six years ago. Finally, to corroborate that brain changes detected were related to motherhood, we examined whether brain changes between the PRE and POST6y sessions predicted the measures of mother-to-infant attachment collected during the early postpartum.

## 2. Materials and Methods

### 2.1. Participants

In the previous study, we followed a prospective cohort study of 25 first-time mothers and 20 nulliparous women that was set up to examine the effects of pregnancy on the human brain [14]. For first-time mothers, the data set included one MRI session before conception (PRE) and another during the early postpartum (POST). Mean time between the PRE and POST sessions was 1.27 (±0.30) years. For nulliparous women, two MRI sessions were acquired at comparable time intervals (Table 1).

To assess the long-term effects of pregnancy, we re-contacted the participants for a new MRI scanning session six years after parturition (POST6y) [14]. Among the 25 mothers included in the original sample, five already had another child, one was pregnant, one had a traumatic brain injury, three had moved out of the country, and eight either did not reply or were no longer interested in participating in the study. This resulted in a final sample of seven mothers (mean age 40.55 ± (3.14) years). For these seven mothers, we also recovered their scores on the Maternal Postnatal Attachment Scale (MPAS), administered during the first months of postpartum. Mean values for the three MPAS subscales were as follows: “Quality of Attachment” = 36.87 (±4.75), “Absence of Hostility” = 16.19 (±2.94), and “Pleasure in Interaction” = 22.00 (±1.83). To control for the effects of other unknown confounding variables, we also contacted the nulliparous women of the original sample of Hoekzema et al., 2017 [14]. Among the 20 nulliparous women, 14 had become mothers, and three were no longer interested in participating in the study. To increase the sample, for the present study, we also scanned two nulliparous women who were not included in the previous study, but whose anatomical scans were obtained at comparable dates, in the same scanner and using identical sequence parameters. For one of these women, a session matching the POST session time interval was also available, resulting in a sample of five nulliparous women (mean age 38.83 ± (6.37) years).

Informed consent for the six-year follow-up session was obtained from all participants. Table 1 provides a general description of the participants at each session. For further details about the original sample and the analyses investigating the impact of demographic or clinical factors on the results, see Hoekzema et al., 2017 [14] and Appendix A.

### 2.2. MRI Data Acquisition

High-resolution anatomical MRI brain scans were acquired in a Philips Achieva 3 Tesla scanner located at Hospital del Mar in Barcelona. The protocol was identical to that of the previous sessions [15], consisting of a T1-weighted gradient echo pulse sequence (repetition time = 8.2 milliseconds, echo time = 3.7 milliseconds, voxel size = 0.9375 mm × 0.9375 mm × 1 mm, field of view = 240 mm × 240 mm × 180 mm, and flip angle = 8°). During the POST6y session acquisition, there was a software update on the Philips scanner (from version 5.1.7 to version 5.3.1), and six participants out of 12 were scanned with a different software version. The effects caused by the software update were isolated and corrected before the image processing (see Appendix A for the correction details).

### 2.3. Image Processing

Images were processed with SPM12 (https://www.fil.ion.ucl.ac.uk/spm/), implemented in Matlab 2017b (https://es.mathworks.com/). We used the longitudinal symmetric diffeomorphic modeling pipeline [20], which produces for each subject a map of volume change between sessions. We calculated the maps of volume change between PRE and POST, and between PRE and POST6y sessions.

As previously mentioned, this study was a follow-up analysis, built upon the results obtained in a previously published paper [20]. Thus, to avoid any potential bias due to image processing steps, we applied a pipeline identical to that used in the previous study (see Appendix A) [20]. Specifically, the anatomical images of each participant were longitudinally registered. This step included rigid-body registration, intensity inhomogeneity correction, and nonlinear diffeomorphic registration. The subjects’ images were then registered to the within-subject average image to avoid biases associated with asymmetry in pairwise registration, giving as a result the Jacobian determinants of the longitudinal pairwise registration. The midpoint average images were segmented into specific tissues using the unified segmentation algorithm [21]. Jacobian determinants were subsequently multiplied by each subject’s GM segmentation, creating maps of GM volume change. These maps of GM volume change were normalized into Montreal Neurological Institute (MNI) space using Diffeomorphic Anatomical Registration Through Exponentiated Lie algebra (DARTEL) tools and smoothed with a 12-mm, full-width, half-maximum smoothing kernel [22].

### 2.4. Statistical Analyses

#### 2.4.1. Region of Interest Analyses

We performed a region of interest (ROI) analysis to focus on the specific areas known to be affected during pregnancy. These ROIs were obtained from the results reported in Hoekzema et al., 2017 [14], and correspond to regions where GM volume decreases more in the mothers than in nulliparous women (GM volume changes ((Nulliparous POST-PRE) − (Mothers POST-PRE))). Those ROIs, named based on their location within the MNI space, are: Left Fusiform, Left Hippocampus, Left Inferior Frontal, Left Inferior Orbitofrontal, Left Middle Frontal, Left Superior Temporal, Medial Frontal, Precuneus, Right Fusiform, Right Inferior Frontal, and Right Superior Temporal (Appendix A). We created an additional ROI including all the above mentioned regions named “All ROIs.”

For each ROI, we tested whether the mean GM volume changes (POST6y-PRE) and the slopes (PRE to POST, and POST to POST6y) differed between groups after correcting for the potential age effects. We used the non-parametric Wilcoxon–Mann–Whitney test to determine the significance of the group comparisons. The test was limited to one side as we specifically expected to find more decreases in the mothers than in the nulliparous group. The threshold was set to False Discovery Rate-Adjusted *p*-value (*q*-value) of 0.05 to correct for multiple comparisons [23].

#### 2.4.2. Whole-Brain Analyses

As exploratory analyses, we examined if groups showed GM volume increases or decreases (between the PRE and POST6y) in other regions not included within the ROIs. We did a whole-brain, voxel-based analysis through a General Linear Model, including age as a covariate. The significance threshold was set to *p*-value < 0.05 family-wise error-corrected, and a cluster of 25 contiguous voxels.

#### 2.4.3. Pattern Recognition Analyses

For classification and regression tests, we performed the multivariate pattern recognition analyses in PRoNTo (version 2.10; http://www.mlnl.cs.ucl.ac.uk/pronto/), implemented in Matlab. Briefly, the pipeline searches for regularities in the maps of GM volume changes and trains a decision function. This function is used to predict the label of the images based on the signal regional contribution within the image. When the labels are discrete (e.g., mothers vs. nulliparous women), the learned function is called the classifier model; when they are continuous (e.g., MPAS scores), it is called regression model. Here, we performed both approaches. First, we used a support vector machine model to test the power of the GM changes (POST6y-PRE) to discriminate between the groups while including age at POST6y as a potential confounder. Then, with a multiple kernel ridge regression algorithm, we tested whether mothers’ brain changes could predict MPAS scores. To evaluate the performance of the pattern recognition analyses, we examined the accuracy of the classification and the goodness of fit measures for the regression models using a leave-one-out, cross-validation strategy. The models’ statistical significances were calculated non-parametrically with 10,000 permutations at a threshold of *p*-value < 0.05.

#### 2.4.4. Positive Predictive Value (PPV)

Given our limited sample size at POST6Y, as a post hoc analysis, we calculated the positive predictive value (PPV) of our results [24,25]. The PPV is the probability that a “positive” or significant finding reflects a true effect. This probability was calculated for the between-group differences in GM volume change (POST6y-PRE) in the “All ROIs” mask (Appendix A).

#### 2.4.5. Supplementary Analyses

Due to an unexpected technical problem, the radiofrequency head coil (RFHC) had to be replaced for three (two nulliparous and one primiparous woman) participants out of 12. To ensure that our findings did not depend on this variable, we repeated the main analysis excluding these participants. Appendix A show that the main findings persisted after excluding the subjects with a different RFHC.

## 3. Results

### 3.1. Region of Interest Analyses

When comparing the GM volume changes between the PRE and POST6y in the “All ROIs” mask, we found larger decreases in the mothers’ group than in the nulliparous’ group (*p*-value = 0.015). Figure 1 and Appendix A show the estimated GM volume trajectories (means and slopes) in both groups, for every ROI and every session (PRE, POST, and POST6y). Results of the early postpartum session belonged to the previous study of Hoekzema et al., 2017 [14], and are included in Figure 1 as reference. As can be observed, most of the ROIs followed the same pattern of change and remained reduced six years after parturition (POST6y-PRE). When studying the group differences at POST6y, all the ROIs survived multiple comparisons, except for the Left Hippocampus (*p*-value = 0.053), Left Inferior Orbitofrontal (*p*-value = 0.074), and Left Superior Temporal (*p*-value = 0.053) that did not reach the threshold established for statistical significance.

### 3.2. Whole-Brain Analyses

Whole-brain, voxel-wise comparisons also indicated that prominent GM volume decreased in the mothers’ group. As shown in Figure 2, these volume decreases affected regions within the explored ROIs, especially those of the medial wall such as the Medial Prefrontal cortex, the Precuneus, and other lateral areas outside the ROIs. We did not find any significant increase in the mothers’ group or changes—either increases or decreases—within the nulliparous women.

### 3.3. Pattern Recognition Analyses

#### 3.3.1. Classification

Figure 3 shows the results of the classification analysis. At six years after parturition, it is still possible to accurately discriminate between mothers and nulliparous women. Based on the GM volume changes between the PRE and the POST6y session, all mothers were correctly identified (class accuracy = 100%, *p*-value = 0.023), and only one nulliparous woman was misclassified as a mother (class accuracy = 80%, *p*-value = 0.059). This resulted in a total accuracy of 91.67% of women correctly classified and a group balanced accuracy of 90% (*p*-value ≤ 0.011).

#### 3.3.2. Regression

Regression analyses indicated that GM volume changes between the PRE and POST6y sessions significantly predicted postpartum scores of the MPAS subscale “Pleasure in Interaction” (Figure 4). Post hoc Spearman correlations indicated that the larger GM reductions at POST6y, the higher scores on the scale “Pleasure in Interaction” (Figure 5). No significant predictions were found for the MPAS subscales of “Absence of Hostility” and “Quality of Attachment”.

### 3.4. Positive Predictive Value (PPV)

The estimated Cohen’s d of the between-group differences in the GM volume change (POST6-PRE) of the “All ROIs” mask was 1.635, and the statistical power of the test was 0.768 (Appendix A). The large effect size counteracted the reduced PPV commonly associated with the reduced sample size. As indicated in Appendix A, the probability that the between-group difference in the “All ROIs” mask reflected a true effect, that is, the PPV, was 0.939.

## 4. Discussion

We found that most pregnancy-induced GM volume reductions in ToM brain regions persist at least six years after parturition. Based on GM volume changes at six years postpartum, we can classify women as having been pregnant or not with 91.67% of total accuracy. We also found that GM brain changes six years after parturition are associated with measurements of mother-to-infant attachment collected during early postpartum, supporting the hypothesis that the detected brain changes between the PRE and POST6y sessions are, indeed, related to early postpartum scores on the maternal attachment scale.

To put in perspective the following discussion, we would like to first address the main limitation of our study: the reduced sample size at POST6y. Small sample sizes are common in longitudinal studies tracking the effect of pregnancy on the brain. Aside from Hoekzema et al., 2017 [14], the other published study describing the impact of pregnancy on the human brain included data from only nine mothers, only two of whom had pre-pregnancy scans [13]. The reason behind such limitation is that scanning participants before and after pregnancy is challenging because it implies predicting the moment of conception and ensuring the viability of the pregnancies. For the current study, which was a continuation of Hoekzema et al., 2017 [14], we started with a sample of 25 primiparous women and followed them for six years. For the six-year, follow-up MRI session, we had to exclude mothers with second pregnancies, as well as female controls who became mothers. This, in addition to the typical dropout of such long longitudinal studies, led to a final sample of seven first-time mothers and five nulliparous women. To date, this sample represents the only longitudinal data set available to examine the long-term effects of pregnancy on the human brain. Statistically, we tried to minimize the caveats of small sample sizes by using more robust non-parametric tests and selecting a restrictive threshold corrected for multiple comparisons. Also, to determine the ratio of true “positive” findings, we calculated the PPV of our results. As our main effects were large, the PPV associated was also large. Specifically, the estimated probability that our significant findings indeed reflected a true effect was 0.939.

Despite the abovementioned sample size limitations, the insights provided from this unique data set indicate that pregnancy leads to GM volume reductions that are still detectable six years after parturition. This finding supports previous literature and provides new insights about the trajectories and the temporality of the brain changes accompanying motherhood. Before the current study, only six longitudinal MRI studies were designed to test the impact of motherhood on the anatomy of a woman’s brain [13,15,17,26]. Whereas Kim et al., 2010 [16], Lisofsky et al., 2019 [17], and Luders et al., 2018 [26], approached the topic of motherhood by analyzing how the brain changes during the postpartum period, only Oatridge et al., 2002 [13], and Hoekzema et al., 2017 [14], included the pregnancy period. Together, these studies suggest that there are GM volume decreases during pregnancy [13,14], followed by increases after parturition [13,17,26].

There is controversy on whether GM reductions fully recover during the postpartum. According to Oatridge et al., 2002 [13], there is a reduction in brain size during pregnancy with an inflection point at parturition and a recovery at six months postpartum. On the contrary, Hoekzema et al., 2017 [14], found that the GM volume reductions detected during pregnancy persisted at least two years after parturition. As opposed to Oatridge et al., 2002 [13], in our previous study we included a control group and restricted the sample to primiparous women [14]. The inclusion of a group of nulliparous women is crucial to control for brain changes induced by aging, as well as by other possibly unknown confounding factors. Besides, it is essential to control for the effect of previous pregnancies since, according to rodent literature, the number of pregnancies affects behavior, neurobiology, and hormonal sensitivity [27]. Here, we showed that most of the GM volume reductions observed when comparing pre-pregnancy with postpartum brains are still present six years after parturition. Likewise, whole-brain analyses revealed that GM volume decreases, while no significant increases with respect to the PRE session were detected. Our results also revealed that women were classified as being mothers or not with a 91.67% total accuracy based on GM volume changes. Every mother was correctly classified, and only one nulliparous woman was misclassified. Altogether, our data suggest that the magnitude of GM volume decreases associated with pregnancy exceeds potential increases during early postpartum and that, even six years after parturition, pregnancy-induced GM volume changes have not remitted. Literature in rodents indicates that some of the behavioral and neural changes induced by pregnancy are maintained after the weaning period [28]. Behaviorally, the dams of post-weaned rats are less anxious and fearful than virgin rats [29] and have better foraging and spatial memory skills [30,31,32,33]. Also, primiparous rats initiate maternal behavior faster when grouped with donors’ pups [28], and multiparous mice retrieve pups faster than primiparous dams [34], thus illustrating the long-term priming effects of previous pregnancies. Neurally, the previous reproductive experience is known to cause long-term changes in crucial areas of the maternal brain circuit, especially in the hippocampus and hypothalamus. Specifically, the hippocampus of post-weaning primiparous rats differs from that of virgin rats in several aspects such as dendritic structure [35,36], amount of neural aging [31], and estrogens’ sensitivity [37]. Besides, in mice, post-weaning primiparous dams exhibit altered hypothalamic gene expression compared to virgin mice [38].

In humans, recent cross-sectional studies also support the long-term effects of parenthood on the brain. In particular, they indicate that elderly subjects who were parents differ anatomically from those who did not have children [39,40]. Those long-term effects may be mediated by hormonal factors. Other periods of acute exposure to estrogens, such as adolescence or hormone therapy in male-to-female transgender subjects, lead to reductions in GM volume that extend beyond the period of hormonal exposure [41,42]. Indeed, some of these GM volume reductions, such as those observed during adolescence, are considered life-lasting. We recently showed that the profile of neuroanatomical changes induced by pregnancy resemble those occurring during adolescence [15]. There, we discussed the potential biological mechanisms behind these GM volume reductions [15]. Here, we showed that at least some pregnancy-induced brain changes remain several years after parturition. Findings open the possibility that brain changes induced by pregnancy are, indeed, permanent.

Besides hormonal factors, another plausible mediator of the enduring effects of motherhood on the brain is the day-to-day caring of the infant. Research has shown that the interaction with the baby during the postpartum period can impact the anatomy and functionality of the caregiver’s brain, either in mothers [16,43], fathers [44,45], or foster parents [46]. The long-term maternal commitment forces women to continuously stay alert for the infants’ needs and to engage multitasking strategies to take care of them. In fact, enhanced working memory has been reported long after the weaning in female rodents that both gestated and mothered the pups, but not in females that had only undergone pregnancy [47], suggesting that some of the brain changes in mothers are long-term maintained by the ongoing relationship with the child.

In addition, it is plausible that factors not directly related to pregnancy and motherhood could account for the observed brain changes. However, we believe this is unlikely. To minimize the possible effect of confounding factors, we used three strategies. First, we restricted the analysis to the ROIs that, according to the results of Hoekzema et al., 2017 [14], underwent a GM volume reduction between the pre-pregnancy and post-pregnancy sessions. Importantly, quantification analyses with all the functional brain networks (seven-network parcellation by Yeo et al., 2011 [48]) showed that these ROIs overlap with regions involved in ToM [14], a brain network that has been extensively related to maternal behavior [49]. Second, we performed a whole-brain analysis of GM volume changes (between the pre-pregnancy and the six years after parturition sessions), in which we included age as a nuisance covariate. The results indicated that the GM volume reductions detected in the current study are not driven by age variability. Finally, we tested the association between the gray matter volume reductions in the “All ROIs” mask observed at six years postpartum and a key feature of maternal care: mother-to-infant attachment. Specifically, we analyzed whether the observed brain changes were related to the scores obtained during the early postpartum on the MPAS, a scale that measures maternal attachment. Both prediction models and correlation analysis suggest that the observed brain changes are related to maternal attachment. Specifically, the more pleasure the mother reported while interacting with her child, the greater the decrease in GM volume between sessions. Hence, although we cannot entirely discard the influence of other unknown factors, a plausible interpretation is that the brain changes found at six years postpartum are related to pregnancy-related factors.

Although more research is needed, our findings are consistent with animal research, which indicates that gray matter modifications are triggered by pregnancy and peripartum hormonal fluctuations and play a critical role in maternal care. Indeed, a recent study suggests that pregnancy might represent another time of a woman’s lifespan akin to puberty, where hormonal fluctuations might have organizational effects on the brain structure [15].

## 5. Conclusions

In conclusion, we used a unique data set to study the long-term effects of pregnancy on the human brain. Our results illustrate that pregnancy-induced brain changes are detectable even at six years after parturition; at this period, the brain of a mother is still different from that of a nulliparous woman. In fact, based exclusively on GM volume changes, we can correctly classify women as having undergone pregnancy or not with 91.67% of total accuracy. Those brain changes seem to be related to maternal behavior, as they predict the measure of mother-to-infant attachment collected during the early postpartum. These findings open the possibility that the brain changes induced by pregnancy are lifelong and enduring.

## Figures and Tables

**Figure 1 brainsci-11-00168-f001:**
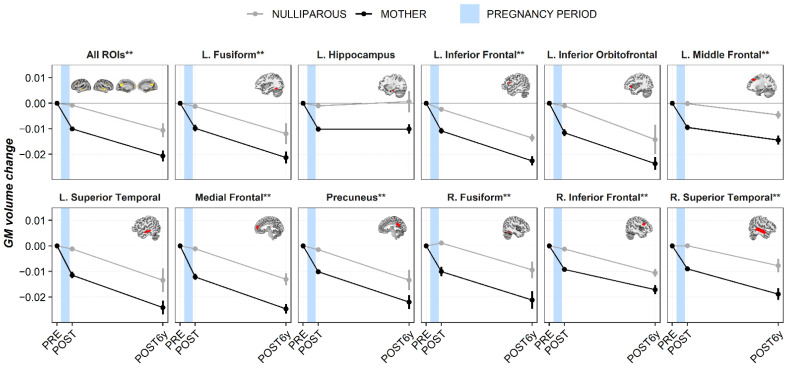
Gray matter volume changes for every region of interest at every session. Results of the early postpartum session were displayed as reference values [14]. Mean values (circle) with their respective standard error of the mean (vertical lines) and slopes (lines joining the circles) are represented. Black and gray lines represent mothers and nulliparous women, respectively. The blue shadow indicates the approximated period of pregnancy. Abbreviations are as follows: GM = gray matter, L. = left hemisphere, R. = right hemisphere, PRE = pre-pregnancy session, POST = early postpartum session, POST6y = six years after parturition session, and FDR = false discovery rate. ** Asterisks indicate group differences at *q* < 0.05 FDR-corrected for multiple comparisons.

**Figure 2 brainsci-11-00168-f002:**
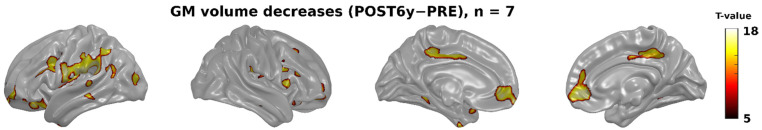
Gray matter volume decreases between the PRE and the POST6y sessions in mothers (*n* = 7), controlling for the effect of age (*p*-value < 0.05, FWE-corrected, and clusters bigger than 25 contiguous voxels). The vertical color bar shows the T statistical values. Abbreviations are as follows: GM = gray matter, PRE = pre-pregnancy session, POST6y = six years after parturition session, FWE = family-wise error.

**Figure 3 brainsci-11-00168-f003:**
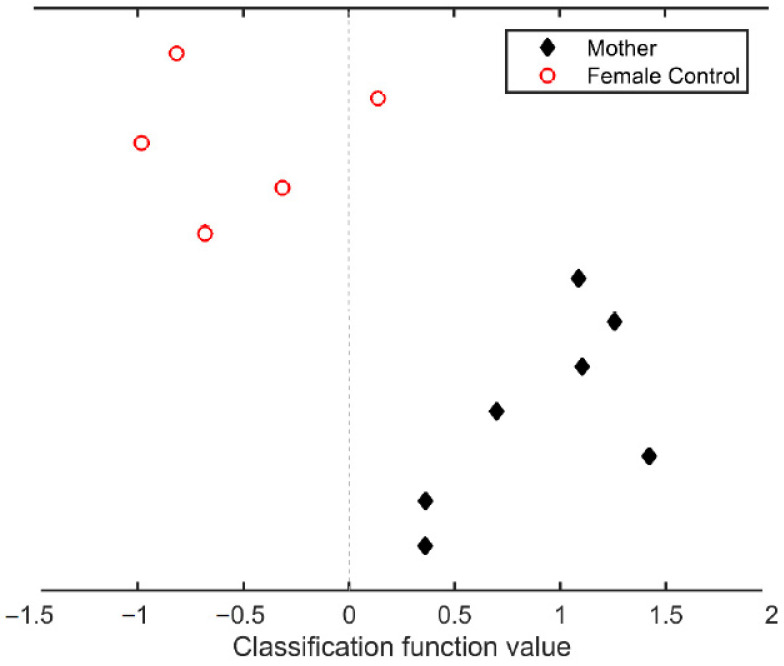
Classification analysis based on the gray matter volume changes between pre-pregnancy and six years after parturition sessions. Black diamonds represent the mothers, and white circles represent the nulliparous women. The dashed line is the cutoff function value between both groups. Classification function values (mean ± s.d.) for mothers and for nulliparous women are 0.899 ± 0.428 and −0.530 ± 0.448, respectively.

**Figure 4 brainsci-11-00168-f004:**
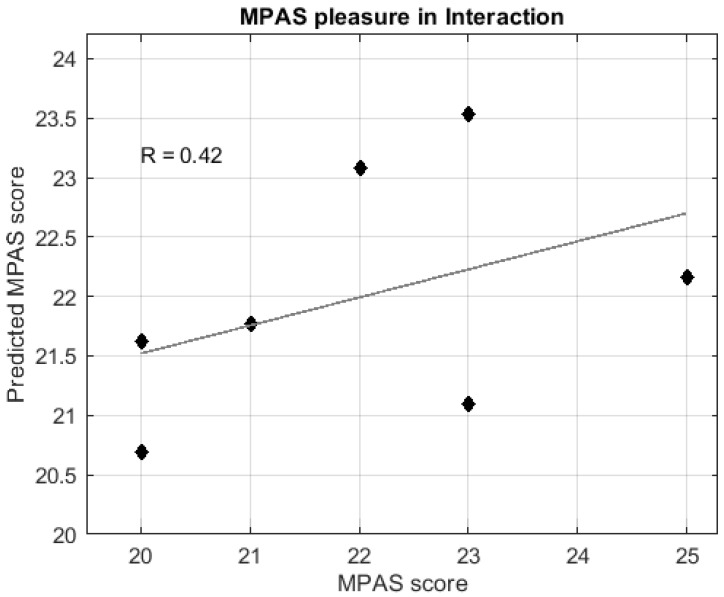
Regression analysis of the postpartum MPAS subscale of “Pleasure in Interaction” based on the gray matter volume changes between pre-pregnancy and six years after parturition sessions. The graph shows predicted (X-axis) versus actual “Pleasure in Interaction” MPAS score (Y-axis). The mean value (±s.d.) of the predicted score is 22.127 (±1.140), the correlation coefficient with the actual score is (R) = 0.65, *p*-value = 0.020, nMSE = 0.330, and pnMSE = 0.017. Abbreviations are as follows: MPAS = maternal postpartum attachment scale, s.d. = standard deviation, nMSE = normalized mean squared error, pnMSE = *p*-value of normalized mean squared error.

**Figure 5 brainsci-11-00168-f005:**
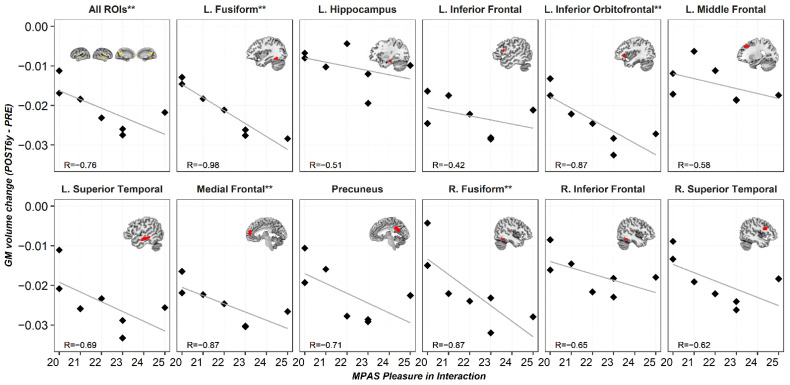
Scatter plots for the Spearman correlations: Y-axis represents gray matter volume changes (POST6y-PRE) for every region of interest, and X-axis represents postpartum scores in the MPAS subscale of “Pleasure in Interaction.” Abbreviations are as follows: L. = left hemisphere, R. = right hemisphere, PRE = pre-pregnancy session, POST6y = six years after parturition session, MPAS = maternal postpartum attachment scale, R = Spearman correlation coefficient, and FDR = false discovery rate. ** Asterisks indicate correlations at *q* < 0.05 FDR-corrected for multiple comparisons.

**Table 1 brainsci-11-00168-t001:** Demographic characteristics of the sample.

Group	MOTHERS			NULLIPAROUS		
Session	**PRE**	**POST**	**POST6y**	**PRE**	**POST**	**POST6y**
*n* of participants	25	25	7	22	21	5
Age (mean ± s.d.) (years)	33.87 ± 3.89	35.14 ± 3.87	40.55 ± 3.14	31.17 ± 5.77	32.12 ± 6.03	38.83± 6.37
Education (*n* of subjects)	2	2	0	2	2	1
-School	4	4	2	4	3	1
-College	19	19	5	16	16	3
-University						
Means of conception (*n* of subjects)	9	9	3	-	-	-
-Natural	16	16	4			
-Fertility assisted						
Time since the PRE session (mean ± s.d.) (years)	-	1.27 ± 0.30	7.22 ± 0.50	-	1.11 ± 0.31	7.57 ± 0.56
Time since parturition date (mean ± s.d.) (months)	-	2.45 ± 1.59	75.80 ± 7.07	-	-	-

The variables “age”, “education”, and “time since the PRE session” did not differ significantly between the groups of mothers and nulliparous women at POST6y (age: *p*-value = 0.545, education: *p*-value = 0.462, time since the PRE session: *p*-value = 0.285). The means of conception at the PRE and POST sessions were skewed toward fertility treatment. However, as stated in Hoekzema et al., 2017 [14], “there were no gray matter volume differences between the groups.” Abbreviations are as follows: PRE = pre-pregnancy session, POST = early postpartum session, POST6y = six years after parturition session, s.d. = standard deviation.

## Data Availability

The data that support the findings of this study are available on request from the corresponding author, Susanna Carmona. The data are not publicly available due to their containing information that could compromise the privacy of research participants.

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
