# Peer review of "Do Pregnancy-Induced Brain Changes Reverse? The Brain of a Mother Six Years after Parturition"

_brainsci, 2021, doi:10.3390/brainsci11020168_

Round 1

Reviewer 1 Report

In this interesting long-term prospective study authors showd total gray matter change in women after pregnancy. The study is consice and provides some valuble information. In my opinion it shoudl be enriched with some clincal data if possible.

  1. Duration and type of labour should be given. Perinatal stress is so larger it can even cause cerebral hemorrhage. It is rational that this might influence cerebral paramteres.
  2. If possible pior disease and medication shoudl be given in separte clinical table. Parameters describing course of pregnancy should also enriche the manuscript.
  3. Were participants of the study screened with MMSE ?
  4. Was any psychological work-up preformed for example to screen for prepartume or postpartume depression ?

Author Response

Thank you for considering a revision of our manuscript. We thank the reviewers for their comments and suggestions. We respond to each of the comments below.

Best,

Susanna Carmona

On behalf of all the co-authors

Reviewer 1:Comments and Suggestions for Authors

In this interesting long-term prospective study authors showd total gray matter change in women after pregnancy. The study is consice and provides some valuble information. In my opinion it shoudl be enriched with some clincal data if possible.

  1. Duration and type of labour should be given. Perinatal stress is so larger it can even cause cerebral hemorrhage. It is rational that this might influence cerebral paramteres.
  2. If possible pior disease and medication shoudl be given in separte clinical table. Parameters describing course of pregnancy should also enriche the manuscript.
  3. Were participants of the study screened with MMSE ?
  4. Was any psychological work-up preformed for example to screen for prepartume or postpartume depression ?

Response to Reviewer 1:

We agree with the reviewer that clinical and cognitive information is relevant to contextualize our findings. Following the reviewer’s suggestions, we have included two additional tables in the supplementary material. Supplementary Table 1 describes clinical information during pregnancy and at the postpartum period, and labor-related information (duration and type of parturition, feeding method, and number of fetuses). Supplementary Table 2 describes pre to postpartum changes in cognitive measures such as learning abilities, working memory, and reaction time.

The previous study of Hoekzema et al. 2017 already investigated the influence of some of these variables on the model, suggesting that these factors were not driving the observed neural changes. Specifically, very similar results were obtained when 1) including the type of conception, delivery, feeding method, and the number of fetuses as covariates in the model (Supplementary Table 20 of Hoekzema et al. 2017), and 2) excluding from the analysis those mothers who had pregnancy or delivery complications and those who developed postpartum depression (Supplementary Table 19 and 21 of Hoekzema et al. 2017. Besides, correlational analyses revealed no significant associations between the mothers’ gray matter volume changes and cognitive results (Supplementary Table 10 of Hoekzema et al 2017). Finally, as observed in the Supplementary Tables, none of the participants had clinical signs of cerebral hemorrhage during parturition, and there were no significant between-group changes in cognitive performance.

We have now included this information on the footnotes of the Supplementary Tables 1 and 2. We have referred to it in the main manuscript: “For further details about the original sample and the analyses investigating the impact of demographic or clinical factors on the results, see Hoekzema et al. 2017 [14] and Supplementary Tables 1 and 2.” (lines 130 to 133 in the reviewed manuscript).

Reviewer 2 Report

The paper focus on whether the pregnancy-induced brain changes is reversible. The paper is well-written. The results and discussions support the conclusion of the paper. I support the paper and recommend for a minor revision.

(1) My concern is that the methodology part of the paper, especially section 2.3 and 2.4, is not sufficient enough to demonstrate the advantage of the method used in the paper comparing to existing studies. The author should expand these sections to include more information on the methodology.

(2) The author should also demonstrate the reason to choose multiple kernel ridge regression algorithm over other regression methods (e.g. multiple regression method) or Artificial neural network. Is it because of higher accuracy or lower computational time?

Author Response

Thank you for considering a revision of our manuscript. We thank the reviewers for their comments and suggestions. We respond to each of the comments below.

Best,

Susanna Carmona

On behalf of all the co-authors

Reviewer 2: Comments and Suggestions for Authors

The paper focus on whether the pregnancy-induced brain changes is reversible. The paper is well-written. The results and discussions support the conclusion of the paper. I support the paper and recommend for a minor revision.

  • My concern is that the methodology part of the paper, especially section 2.3 and 2.4, is not sufficient enough to demonstrate the advantage of the method used in the paper comparing to existing studies. The author should expand these sections to include more information on the methodology.

(2) The author should also demonstrate the reason to choose multiple kernel ridge regression algorithm over other regression methods (e.g. multiple regression method) or Artificial neural network. Is it because of higher accuracy or lower computational time?

Response to Reviewer 2:

Following the reviewer’s suggestion, we have extended the information provided in sections 2.3 and 2.4 by adding a figure that depicts the processing pipeline and the statistical analyses (see figure in supplementary material “figure S2”).

However, we are unsure what the reviewer means when he/she mentions “existing studies.” We believe he/she refers to voxel-based methods versus surface-based methods, but please, let us know if we have misunderstood her/his comment.

Voxel-based and surface methods are both valid and commonly used methods to assess brain structural longitudinal changes. In this manuscript, the main reason for using the voxel-based morphometric method was to minimize any potential biases induced by using a different processing pipeline to that used in the previous study of Hoekzema et al., 2017. As this is a follow-up study built upon previously published results, the use of a different methodology would suppose a limitation on the interpretation of the results.

The same response applies to the second comment of the reviewer. Regarding this point, we want to clarify further that we have chosen multiple kernel ridge regression for the multivariate pattern regression analysis since it is the most common deterministic method available in PRoNTo. As compared to probabilistic methods, the use of deterministic methods allows for a more direct and clear interpretation of the results. On the other hand, multiple regression is not suitable for this analysis due to the very large dimensionality of the data compared to the number of subjects, which requires the usage of kernel methods and regularization methods to prevent overfitting and, therefore, a lack of accuracy. Also, the use of an Artificial Neural Network would require a much larger amount of data to train the network.

Please, let us know if you think we still need to expand those sections or if we have misunderstood your comment.

Round 2

Reviewer 1 Report

I read improved manuscript with pleasure. It is substantially improved. All the clinical data were given in new concise table that very clearly gives important data. Requested statements for the labour were also presented. References were also improved. I recognize the revision.